

# Failure modes of loose landslide deposits in 2008 Wenchuan earthquake area in China

Jianjun Gan[1,2], Yi Xia Zhang[2]

[1]National-Local Joint Engineering Laboratory of Water Engineering Safety and Efficient Utilization of Resources in Poyang Lake Watershed. Nanchang Institute of Technology, Nanchang, 330099, Jiangxi, China
[2]School of Computing, Engineering and Mathematics, Western Sydney University, 2751, NSW,
Australia.
*Correspondence to: Jianjun Gan (ganjianjun@nit.edu.cn)*

**Abstract:** In this study, a geological investigation and statistical analysis of the post-earthquake slope deposit failures in a meizoseismal area were presented, with a selected example from the 2008 Ms 8.0

Wenchuan earthquake occurred in Sichuan Province in China. The typical slope deposit failures were surveyed in three meizoseismal areas, namely Qingchuan county in Guangyuan city, Beichuan county in Mianyang city, and the epicenter area, Wenchuan county in Aba Tibetan Autonomous Prefecture. According to the movement, material and deformation mechanism of rock or soil, the failures of the post-earthquake landslide deposit could be subdivided into four categories, i.e. slide, collapse, erosion

and flow. This classification of failures of landslide deposit considers the topographic and failure after the earthquake. Besides, some other important factors such as topography, lithology and hydrogeology are also considered. The above mentioned four failure categories are further split into 12 sub-classification. The complicated deformation mechanism and different failure patterns of the slope deposits are analyzed in typical deposits. This classification provides a good reference for the

prediction of geological hazards, whereas mitigation of the landslide or debris flows caused by loose deposits in the meizoseismal area is still a difficult task.

## 1. Introduction

The failure types of post-earthquake deposits have been examined in several studies, and classification (1938) is primarily based on materials (earth and rock), movement (flow and slip) and velocity (slow

or very rapid), without considering the effect of topography, landform, volume and inducing mechanism. Based on the material and type of movement, Varnes (1954,1978) classified the slope failure into five types, i.e. fall, topple, slide, spread and flow, and this has been the most widely used classification for landslides in the world. According to seismic parameters, materials and geologic environment, Keefer (1984) divided the landslides into 14 types. Considering the landslide shape and



geotechnical parameters, Hutchinson (1988) divided the slope deformation failure modes into creep, frozen ground phenomena and landslides, but did not consider the trigger mechanism and the effect of the volume. Hungr (2001) classified the landslides into ten types based on the genetic and morphological characteristics, which introduced a new category in combination with unsorted material and sorted material. However, the deformation-failure modes, particularity of the loose

post-earthquake main body has not been paid much attention in previous studies, and further studies should be conducted based on these landslide classifications.

    The purpose of the new classification propose in this paper is that landslide deposits can be effectively split into common categories according to deformation mechanisms, which retains the established concept and reveals the deformation and failure trend of landslide events. This is easier to achieve

with a statistical analysis of a field survey, without resorting to more complex taxonomic methods. Moreover, understanding deformation and failure mode could help to mitigate and prevent similar geological disasters. Some authors have made good attempts and achieved significant results. For instance, the "locking section" was used in one study of mechanisms of large-scale landslides occurred in China by Huang (2011) to identifyy a three-section model which includes sliding, tension

cracking and shearing. Using the same apparatus, Yang (2015) also evaluated the post-earthquake rainfall-triggered deposit failure occurred in Lushan area, Sichuan province, China.

    The discussion of this paper focuses on the deformation and failure mechanism of loose deposits after the earthquake. Although deformation and damage mechanism of the accumulation body have been preliminarily considered, classification and specialty of the landslide deposits have not been well

developed. Wang (1981) found that the after-shock caused the cyclic shear induce the decrease in the strength of sliding surface shear on rock slope instability. Some researchers used inertia, damping, weakening, and liquefied instability to interpret the instability of the deposit. Seed and Martin (1966) used the regular soil deposit for a laboratory test, with limited effort focused on the large deformation of inclined slope caused by material liquefaction. Kramer (1997) proposed that instability of

post-earthquake can be spilt into weakened instability and inertial instability. Based on indoor experiments and field tests, a few researchers studied the liquefaction mechanism and shear deformation of loose deposits after earthquakes in China, Japan, and New Zealand. It was confirmed that liquefaction or shear forces established slope deformation. However, the empirical models for the deformation and failure of loose deposits after such earthquakes have not been proposed.

Nearly 45,000 loose deposits were induced by the 2008 Wenchuan8.0's earthquake in China spreading in 51 disaster areas of 130 thousand $km^2$. These include 13,229 landslide deposits, 5,180 collapse deposits, and 2,400 debris-flow deposits in Sichuan Province, according to the post-earthquake survey (Huang, 2009). Many loose deposits of the Sichuan Post-earthquake areas are susceptible to rainfall or landslides induced by aftershock. More than 12,000 potential geological hazards were triggered by

rainfalls (Fig.1), which killed hundreds of people (Kirschbaum, 2010; Liao, 2011).



A clear classification system of the deformation mode of the accumulation body is more beneficial to the stability evaluation of multifarious geo-hazards. In particular, the geological hazard classification system in strong earthquake areas should consider the effect of multiple factors, such as topography, stratum lithology, material, motion velocity, deformation and failure mechanism. A practical type
classification based on selected attributes is a good classification and a quick way to solve practical engineering problems. According to the material and sedimentological characteristics, Fan (2017) divided the dam landslides caused by the 2008 Wenchuan earthquake into three categories, which will help the prevention and control of landslide dams in strong earthquake areas, however, there is no classification for loose deposits such as debris flows and collapsed deposits.

In this study, the geological conditions and the type of geo-hazards induced by the 2008 Wenchuan earthquake are firstly introduced. Subsequently, the classification method and the typical failure mode of the loose deposits occurred since 2008 are discussed. A new classification method for deformation and failure modes of deposits considering various factors such as topography, material, motion velocity, volume, and particle composition is proposed. The formation mechanism and failure modes
of the geological disasters induced by 12 loose deposits are analyzed. The proposed new classification of failure modes for loose deposits should also be easily applied to the classification of geological hazards occurred in other strong earthquake zones.

## 2. Site Study

### 2.1 Geological conditions


Detailed analyses of the landslide deposits show that the slope deposit failure of the post-earthquake regions in Wenchuan, China are complex. It is important to study the geological conditions in order to recognize the potential geological hazards. The specific failure mode is related to the specific topography, deformation and structure of the rock (soil). This study area has crossed various
geomorphic units, covering Qinghai-Tibet plateau, Longmen mountain, Sichuan basin and valley throughout north to south. The terrain shows high in north and west, but low in south and east. Due to well-developed faults, complicated topography, and various types of rock-soil mass structure and climate change in this area, many post-earthquake loose deposit slopes were accumulated in the potential geo-hazard regions, and it is important to study the failure mode and evolution process of the
Wenchuan earthquake area.



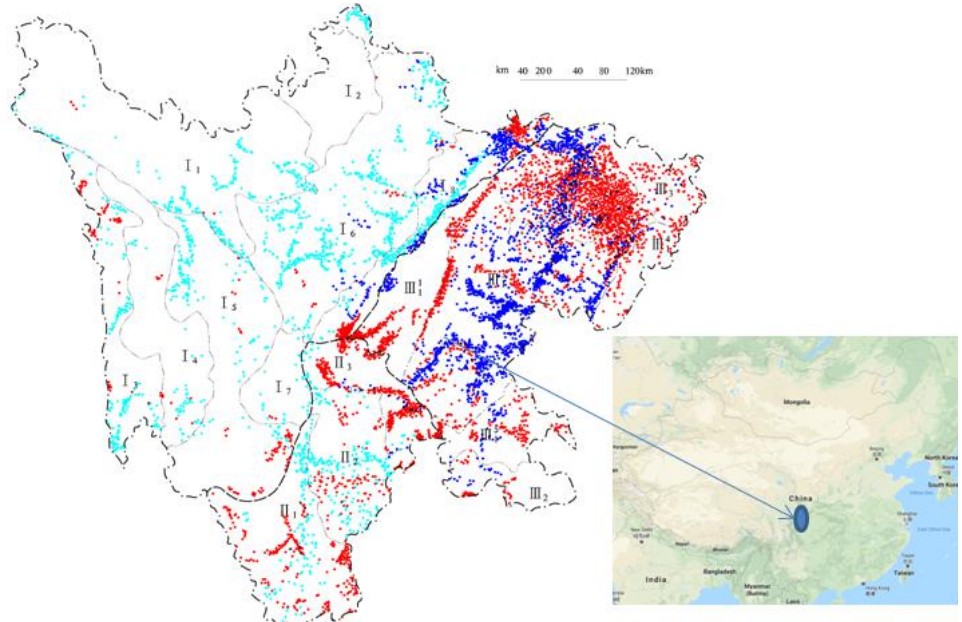

**Fig. 1 Loose Deposits triggered by 2008 Wenchuan 8.0's earthquake in Sichuan Province, China .** (Landslide
deposits are shown in red; collapse deposits are shown in blue; debris flow deposit shown in green. It is included on
13229 landslides, 5180 collapses, and 2400 debris-flows in the study area, by the geological survey in 2009. )

Legend: 🔴 landslide deposit;  🔵 collapse deposit;  🟢 debris flow deposit

I: Hig montain plateau region of western si chuan;I1: Shiqu Seda straucture denued hilly palteau are;I2: Hongyuan Ruoergai tectonic denuded

swampy plains;I3: East bank of Jinsha reive tectonic erosion mountain canyon area;I4: Shapuli moutain ersion or denudation hilly plateau area;

I5: Yalong river strucure erodes the deep valley mountain area;I6: Qionlai moutain to Minshan moutain tectonic erosion ridge mountains;I7:

Gongga mountain structure erodes extremely high mountains;I8: Longmen mountain fault erosion slope in the mountain area.

II Moutain area of southwest SichuanII1: Emei mountain to Wuzhi mount tectonic erosion block mountain area;II2: Xichang Yanyuan

Tectonics erodes moddle mountainous area of wide valley basin;II3: Liangshan tectonic erosion middle mount area.

III: Mount area in eastern basin in Sichuan;III1: Tectonic erosion low mountain hilly in Sichuan Basin; $III_1^1$ : Inclined plain sub-region in the

front of western fault depression basin; $III_1^2$ : Monoclinic low mountain subregion north of tectonic erosion basin; $III_1^3$ : Table low hilly

sub-region south of erosion tectonic basin; $III_1^4$ : Parallelism (low mount) valley (hilly)sub-region in eastern of erosion tectonic basin;III2:

Michang mount to Dab mount tectonic corrosion bedded middle area;III3: Wu mount to Dalou mount stong karst valley middle mountain area.

## 2.2 Seismicity and Rainfall

Several high magnitude earthquake has been recorded in the Longmen Mountain tectonic zone along

the eastern margin of Tibetan Plateau (China) in the last few decades. The Ms7.5 Diexi earthquake on

August 25, 1933, caused the catastrophic landslide which blocked the Minjiang river and formed three

famous "quake lakes". Collapse deposit had slipped into channel and formed landslide dam, then

caused deformation and failure, subsequently the water of this lake pour down, and as a result, 2500

people had been killed and more than 6800 houses had been destroyed (Ren, 2017). The Wenchuan



earthquake on May 12, 2008, and the Lushan earthquake on April 20, 2013, had magnitudes of Ms 8.0
        and Ms 7.0, respectively. These epicenters were located Longmen mountain fault, SW-NE of Chengdu
        City, and the epicenter was located at the depth from 5 to 20 km, within the Eurasian plate of the
        Yangtze plate.

        The above-mentioned earthquakes occurred in the Longmenshan fault zone, indicating that the strong
earthquakes in this area are frequent and the geological environment is very fragile, which is the
        source of power for loose accumulations. These recurring earthquakes are the result of the relative
        uplift of the Tibetan Plateau and the relative decline of the Sichuan Basin. The relative movement of
        the Qinghai-Tibet Plateau and the Sichuan Basin resulted in the uplift of the Longmen Mountains and
        formed a large seismic zone parallel to the eastern margin of the Qinghai-Tibet Plateau. The
Longmenshan fault zone includes three major fractures, namely Maoxian-Wenchuan fault and
        Yingxiu-Beichuan fault and Pengxian-Guanxian fault, which are widely distributed on the two largest
        anticlinorium, i.e. the Pengguan anticlinorium and Baoxing anticlinorium. Due to the violent new
        tectonic movement in the area, the rock mass is broken and the earthquake is frequent, causing a large
        number of loose deposits (Fig. 2). As show in Fig. 2, three major faults were formed, i.e, the fault
located at the junction of Longmen Shan and Sichuan basin, the piedmont fault, also known as
        Pengxian-Guanxian Fault, which is approximately parallel to the Longmen mountain, and the 240 km
        main central fault, which is also known as Yingxiu-Beichuan Fault; and the Maoxian-Wenchuan Fault,
        also known as the Maoxian-Wenchuan Fault.

Most typical loose deposits triggered by the earthquake occurred in Longmen Mountain of Wenchuan,
        which is around 60 km of Chendu city, Sichuan Province, near the east fringe of Tibetan plateau,
        China (Fig.1). Based on the multi-source remote sensing data and field survey data from 2009 to 2018
        provided by the China Geological Survey (CGS), rainfall is the main cause of landslides, collapses,
        and mudslides caused by loose debris deformation. Among them, the period of 2010-2014 is the peak
of the development of rainfall and geo-hazards, and hundreds of geological disasters were caused by
        the failure of loose deposits after the 2008 Wenchuan earthquake.



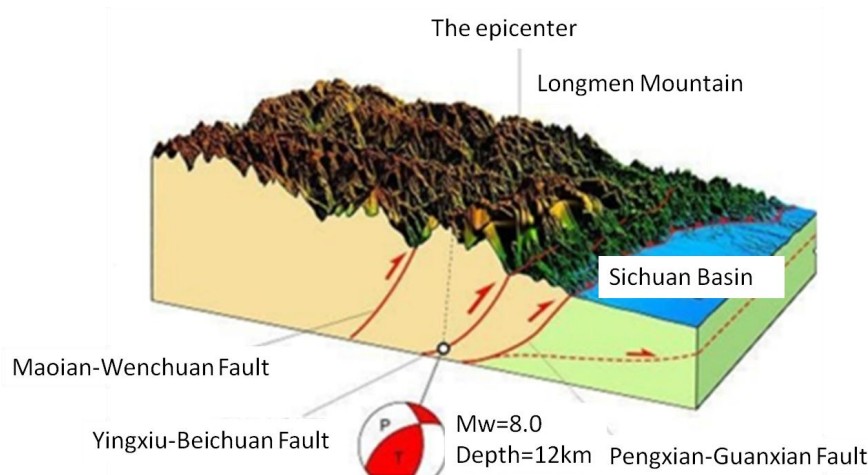

**Fig. 2 Three main faults along Longmen Mountain tectonic.**

It is suspected that the rainfalls and aftershocks have triggered the landslide or debris flow. Rainfall

has played an important role in the conversion of loose accumulations into landslides and has also attracted the attention of many research interests. The study area has a subtropical humid climate and usually brings heavy rainfall between June and September. The average annual precipitation in the study area is $4.87 \times 10^{12}$ m$^3$, and the annual average rainfall is 1003.1 mm. The Longmen mount fault zone is a concentrated rainfall distribution area with a maximum rainfall of 160 mm in 24 hours,

which provides sufficient external dynamic conditions for the loose accumulation failure. In addition, there are more than 1,400 rivers in the study area, and the water flow rate reaches $1.59 \times 10^4$ m$^3$ per second, which is also an important factor for the deformation and failure of loose deposits (Fig. 3). Under the combined action of seismic activity and hydrogeological conditions, the loose accumulation slope in this area has a high risk failure in the earthquake process. These factors must be taken into

consideration in loose deposits failure modes classification.





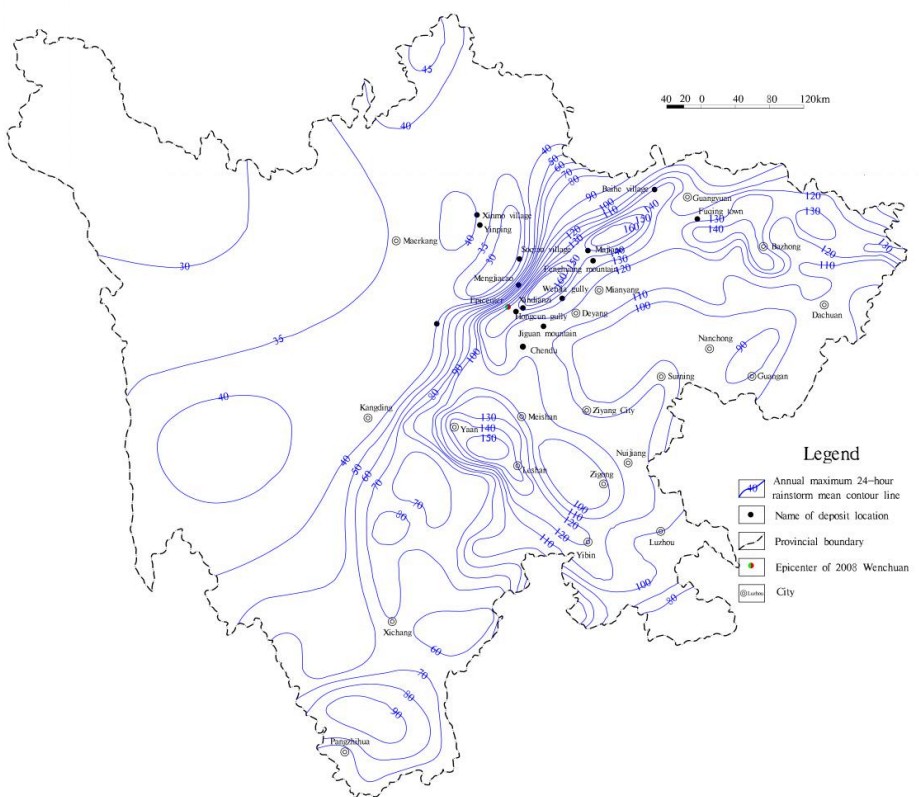

**Fig. 3    24 hours rainfall in Sichuan province**

## 3. Investigation Methods

Field investigations were performed to understand the geological features in the area and the mechanism of the landslides deposits. Methods include outcrop observations and topographical measurements, as well as the use of drilling, trenching and pit exploration to investigate the internal

conditions of loose deposits. Some giant loose deposits also used geological drilling and standard penetration testing (SPT) to study the particle composition. Due to the complex and diverse lithology of the landslide loose deposits, the engineering geological profile of typical loose deposits is drawn based on the investigation and analysis of the lithology of the strata. Finally, the deformation and failure mechanism are analysed. The main field survey site and research object are the most

representative large deposit body within 50 km wide along the Longmen mount fault zone (Fig.3).

The field investigation results reveal that the typical lithology of the deposit is the bedrock which consists of weakly weathered, moderately weathered and strongly weathered coarse and fine granite, limestone and sandstone. Under weathering or post-earthquake weathering, the bedrock is covered



with a large amount of loose clay, broken rock mass or their mixture, which is the main component of
landslide sediments.

According to field investigation statistics for the Wenchuan earthquake area in 2010 (CGS), these
deposits can be classified into four types based on the topographic and type of movement (Cruden
and Varnes,1996), i.e. slide, collapse, erosion, and debris flow type representing for 62.74%,
24.57%, 11.38%, and 1.31% of the deposits, respectively. The ratios of slide, collapse, erosion and
flow type are of 41:29.1:28.6:0.4 in plateau mountain areas. In high to medium mountains in a
transitional zone from plateau to basin, slide of landslide deposits induced by Wenchuan earthquake
is the main failure mode (up to 65.3%), followed by erosion mode with 26.6%, collapse type with
6.5%, and the debris flow with 1.6%. But in basin and mountain area in Sichuan province, the ratios
of slide, collapse, erosion and debris flow type are 66.9:31.1:0.5:1.5 (Table 1).

**Table 1 Category of the landslide deposits in study area**

| Topographic and geomorphic zoning | Type of movement | | | |
|---|---|---|---|---|
| | Slide | Collapse | Erosion | Debris flow |
| Plateau and alpine region | 3105 | 2268 | 2166 | 34 |
| High to medium mountain area | 2311 | 231 | 940 | 57 |
| Basin and mountain area | 8361 | 3886 | 65 | 184 |
| Total number | 13777 | 6385 | 3171 | 275 |

## 4. Typical failure modes of loose deposits

### 4.1 Slide

The slide type of deposits is usually caused by the reconstruction of rock or soil slopes. Under the
action of external geological forces, e.g. rainfall, aftershocks and human engineering activities, the
loose deposits move along the weak surface or sub-surface. According to topography, material
materials, motion characteristics, and on-site investigation, we classify the slide into four categories,
i.e. reactivation of old landslide, slide along the weak soils or rocks, shallow slide of deep deposits and
integral sliding on bedding rock.

### 4.1.1 Reactivation of old landslide

Stable or almost stable ancient landslide deposit body is induced by the earthquake, and subsequently
global or partial reactivation may occur to lead to deformation and failure of accumulation body under
the effects of rainfall conditions, aftershocks and human project activities. For instance,the Xindianzi
landslide, located in Yinxiu town, Wenchuan county, obviously the epicenter of the 2008 Wenchuan
earthquake, is a typical reactivation of old landslide (Fig. 4). The source area of the Xindianzi old
landslide is nearly 0.8 km long and 0.5 km wide, while the old slope angle is $25°\sim30°$. The angle of
old main scarp behind deposits is steep ($45°\sim75°$). The estimated volume of the deposits is $6\times10^6$ m$^3$
and their material is a single and homogeneous, mostly the loose medium granular soil.

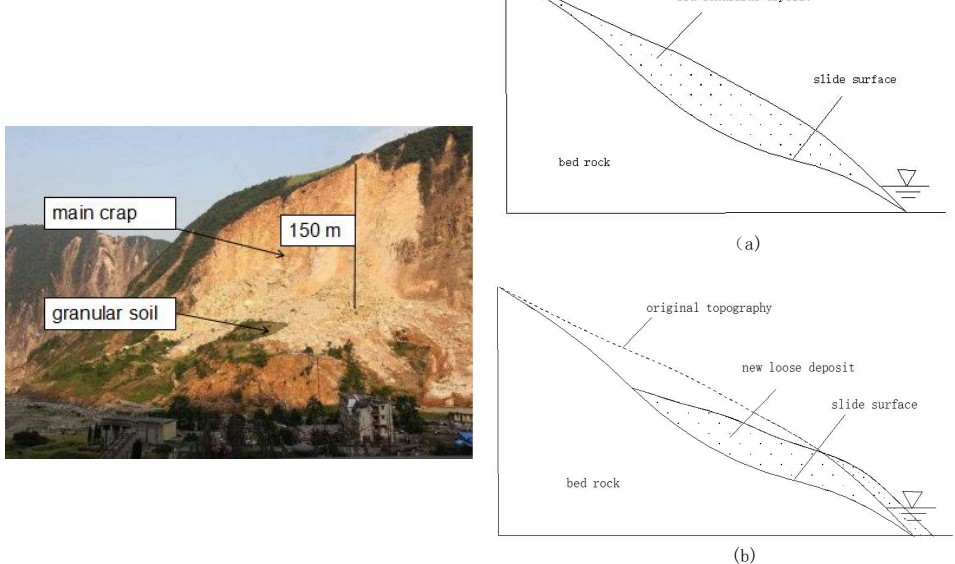

**Fig.4 A slow soils slide on the of Xindianzi old landslide, (a)Schematic of loose deposits before the deformation;**
**(b) Schematic of deposits after the failure, large of homogeneous materials stop in the slope foot.**


Creep and sliding deformation of Xindianzi old landslide is slow at the beginning, but after the strong rainfall infiltration on August 11 2010 and the slope excavation on the crown, the landslide displacement and deformation increased rapidly. The water content of this loose soil accumulations increased rapidly after rainfall, and the gravity of the sliding body also increased. As a result, the shear

strength of the main body composed of loose deposits decreased, and even the strength of the soil decreased resulting in liquefaction. The loose granular structure and high sensitivity to rainwater softening are the basic conditions for the resurrection of ancient landslides, while the most significant localities with extra-sensitive loose deposits are largely distributed around the Yingxiu-Beichuan main fault zone. A large number of reactivations of old landslides have also been found near Tangjia mount

(Hu, 2009)



### 4.1.2 Slide along the weak soils or rocks

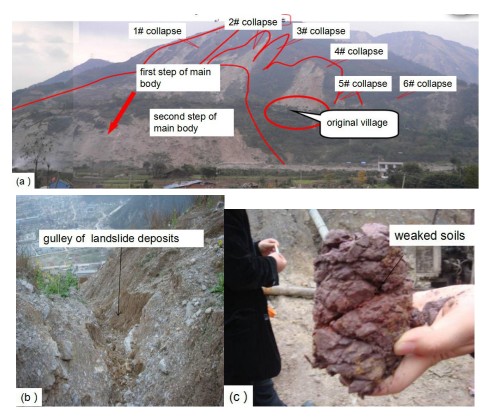
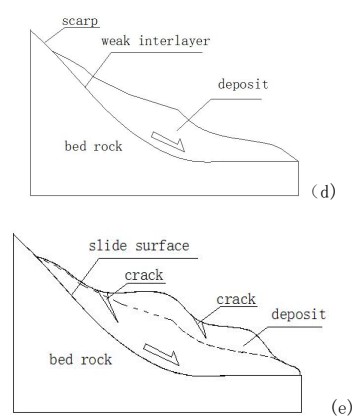

**Fig.5 The landslide occurred at Fenghuang Mountain, Sichuan, China, in 2011.(a)image of Fenghuang Mountain landslide; (b) Gully at the trailing edge of a landslide;(c) Soft crushed soil by drilled;(d) Schematic of loose deposits before failure;(d)Schematic of loose deposits after failure.**

Slides along the weak soils or rocks usually occur in deposits with weak interlayers. The main body consists of loose deposits, broken rocks, and their mixtures. The weak interlayer consists of plastic-soft clay or clastic sediments, and the bedrocks are usually consisted of fully weathered-fully weathered shale, mudstone or sandstone. Before the deformation of the rock and soil in the weak interlayer occurs, the landslide generally moves slowly, and the moving speed is usually less than 0.1m / 1a. Whereas under the influence of earthquakes, rainfall and human engineering activities, the loose deposit will suddenly accelerate in the case of the transfixion of weak interlayer or the weak zone (Huang, 2011).

Fenghuang Mountain landslide locates in Ershe village, Leigu town, Beichuan county, with a total square volume about $1.08 \times 10^6$ m³. It is the slide on the weak interlayer with the following main features: the landslide deposit is nearly 420 m long and 1560 m wide, with the average slope angle of 25°, which is affected by deformation. Its main scarp is 25 m high in average, presenting two moving steps, with the horizontal distance of 167 m and the height of 80 m. The middle of deposits is 111.6 m thick, 94 m thick in the slope toe and 58 m thick in the slope head. Most of the material of this landslide deposits are composed of limestone, carbonaceous shale, silty clay, crushed stone or pebbly clay. The soil sample exposed by drilling is characterized by kneading and water absorption, suggesting that the soil sample is subjected to high compression and grinding. According to geological hazard monitoring, the slip velocity of this accumulation body is 0.08 m/1year. Excavation of the road at the toe of the slope resulted in the rapid down move of the deposit along the weak interlayer (Fig. 5).


### 4.1.3 Shallow slide of deep deposits

A shallow slide on the deep earthquake deposits generally occurs in highly consolidated deep rock and soil. The velocity is extremely high (often greater than 0.1 m/a), and sometimes the surface fragmentation of the soil accelerates with the rise in slope increases throughout the movement. This type of failure is caused by earthquake, rainfall or human activities. It leads to the deterioration of the structure and strength of the shallow surface of the stratum, followed by the creep and sliding deformation of the shallow of deposit body(Fig. 6).

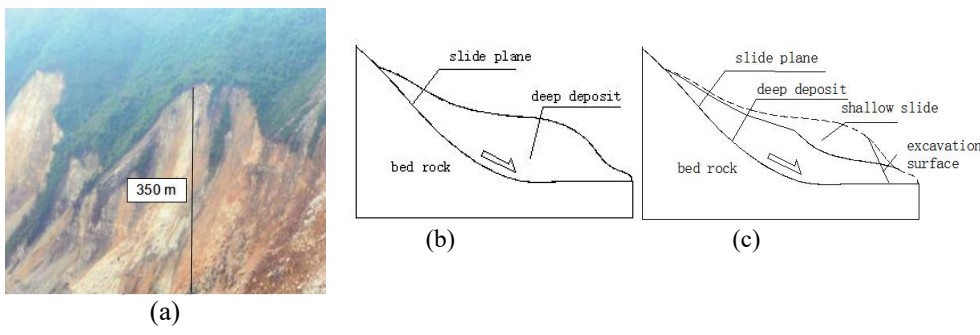

**Fig.6 Majiapo landslide in Beichuan County. (a) Photograph of a shallow slide in Majiapo shallow landslide, Sichuan Province, China;(b) Schematic before failure;(c) Schematic after failure.**


Majiapo landslide locates in Yuli town, Beichuan county, which is nearly 330 m wide and 230 m long. Its volumes is nearly $4 \times 10^5$ m$^3$ and less than 10 m thick of the main body. The landslide deformation was very slow before the Tangjia Mountain earthquake lake was formed. However, after the toe of this deposits was submerged by the water, the shallow landslide moved quickly. The landslide

deposits have a steep ($25° \sim 45°$) slope angle about 28 m high. The source of the deposits is largely composed of gravelly soils with highly weathered phyllite and slate (takes up 50-60%). Likewise, these shallow landslides are known to occur both on the surface land and under the earthquake lake water.

### 4.1.4 Integral sliding on bedding rock

Integral sliding on bedding rock generally occurs in loose rock deposit with a forward gentle laminar rock layer. The topography of this failure mode is characteristic by V-shaped or U-shaped valleys. These slopes are composed of medium-to-sloping layered rocks. They may slide along the bedding plane under the action of their own weight or load, or they may be deformation and failure caused by external loads such as rainfall or earthquakes.





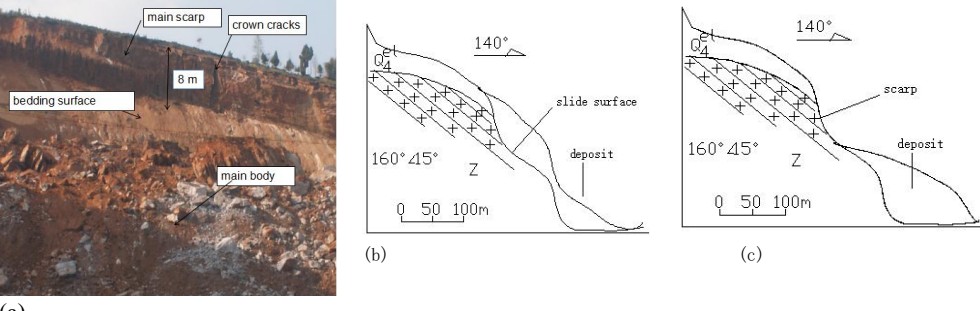

**Fig.7 Photography and schematic of slide in Fuqing town, Wangchang county, Sichuan provice, China,2011.**


An integral landslide is located in Fuqing town, Wangchang county, Sichuan Province. The landslide was formed during the 2008 Wenchuan earthquake as well of which the tectonic crown cracks are 0.5-1.0 cm wide, 1-2 m long, or 0.5-0.8 cm wide, 2-3 m long. The landslide occurred after the constant rain in July 2011, of which the deposit area is $1.36\times10^4\,m^2$ and a total volume of $1.31\times10^5\,m^3$.

The formation lithology in the landslide deposit primarily consists of sandstone of the Triassic system (T) and Quaternary residual slope alluvial soil(Q). The angle of bedding rock is steep (more than 35°), and main body is 9.6 m height in average, of which the main scarp is 8 in height. Remaining unstable landslide height 8 m may slide suddenly in the future. According to field reconnaissance, the velocity of this landslide is a 0.5 m/1 year, and the rainfall infiltration and becomes a weak surface along the

bedding limestone are the main failure factors (Fig.7).

**4.2 Collapse**

Collapse is produced in steep slope deposits which under external forces, including gravity, earthquake, weathering denudation or human activities. It is a single or compound movement with sharp fall, caving, sliding, rolling, jumping, and other special forms, sometimes they hit each other in

the process of movement, then pile in the slope toe (Rens, 2008). Most of the collapse sources are rock deposits with low shear strength and 2-3 groups of penetrating fractures. Whether or not collapse occurs depends on deposits steepness and deposit stability. Based on collapse's travel velocity and movement way, collapse type can be split into the following three sub-types.

**4.2.1 Collapse-slide**

Xinmo catastrophic collapse-sliding rock avalanche is the recent famous massive rock collapse in Wenchuan earthquake area, causing 10 people death and 73 people missing. This massive deposits located in Xinmo village, Diexi town, Mao County, Sichuan province. It may be originated from the 1993 Diexi 7.3 s earthquake that caused several cracks in the slope crown. Besides, after a long period of weathering, rain erosion, and 2008 Wenchuan M8.0 s earthquake, the slops trailing edge fissure




stretched downward and finally passed, and then the massive rock mass traveled more than 2 km. The
        total volume of the rock mass deposits is about $4.5 \times 10^6 \, m^3$, about 210 m long and 300 widths, and the
        fastest traveling velocity of the massive loose landslide deposits is about 74.6 m (Fig. 8) (Xu, 2017;
        Fang, 2017; Meng, 2018).

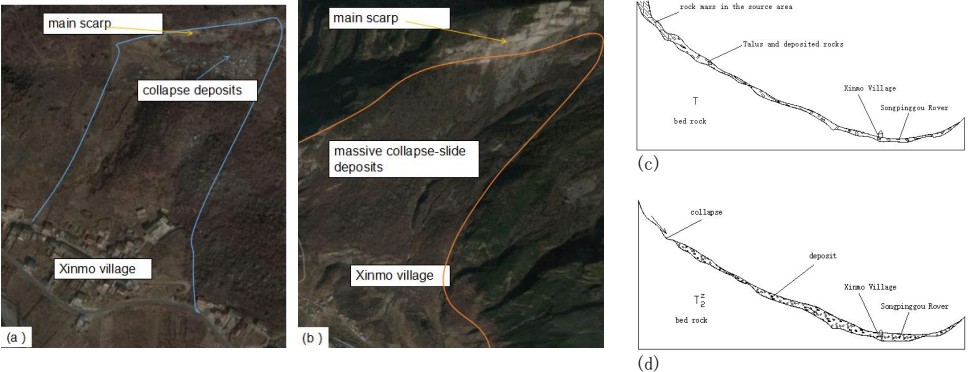

**Fig.8 Photograph and schematic of massive rock collapse in Xinmo village, Diexi Town, Sichuan,2017. (a)photograph of collapse deposits on 10 May 2017; (b) photograph of massive collapse-slide deposits on 20 May 2018; (c) schematic of massive deposit before it's failure; (d) schematic of massive after the failure.**

Massive collapse-sliding is one the catastrophic disasters that pose threats to the people's lives in the
        earthquake area. If the loose deposits consisted of densely structured rocks and joint fissure which had
        an unstable effect on rocks extensive distributed, fractures would be formed through a plane.
        Subsequently, under the action of multiple earthquakes and long-term gravity, the aging deformation
        is generated. When the rainfall accumulates several months and overall the stability of the loose
deposits, the catastrophic landslide may be produced suddenly.

### 4.2.2 Crack-slide collapse

A crack-slide collapse is a form of a steep slope, characterized by steep and vertical fractures on
crown of the slope, occurring when loosely cemented material or rock layers move on a short distance
and dump at the toe of the slope (Tarbuck.1998). Though the surface of the slope displacement is
small, deep crown cracks had been formed by rain infiltrated, earthquake, or weathering (Fig.9).
Moreover, the gravity of overburden deposits based on the weak layer increases in the process of
rainfall, thereby making deposits falls down gradually along a parallel surface. This deformation
mostly occurs in the consequent bedding landslide deposits.



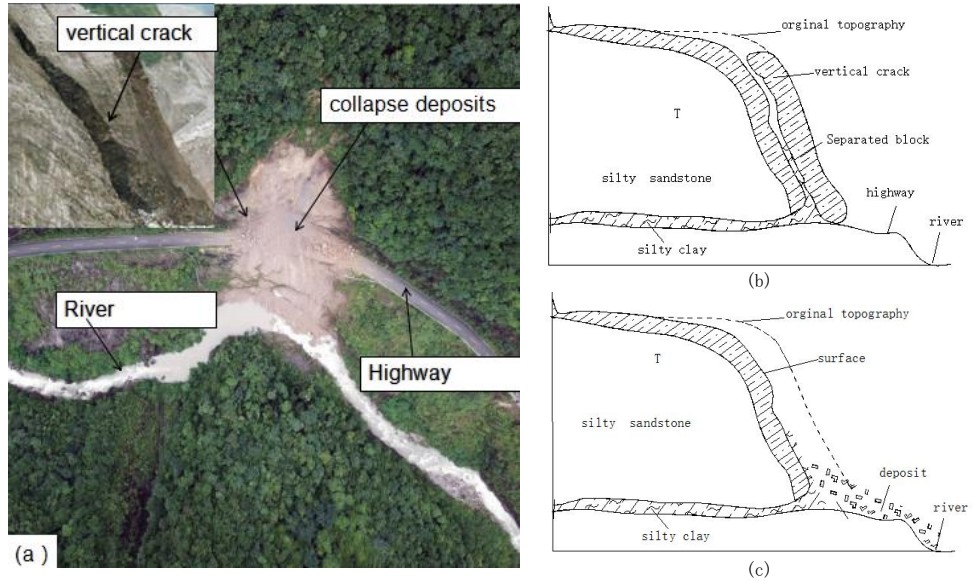

**Fig. 9 Aerial photograph and schematic of crack-slide collapse in Jiguan mountain, Chongzhou City, Sichuan, China,2018. (a)aerial photograph of Jiguan mountain; (b)schematic of loose deposit before failure;(c) schematic of crack-slide collapse.**

Jiguan mountain crack-slide collapse occurred on July 9, 2018, which is about 40 km south of the epicenter of 2008 Wenchuan earthquake. Fig.10 gives an aerial photograph of the collapse. At the crown of the collapse, there were several vertical cracks about 2.5 m deep. The amount of the collapse deposits was about 250 m wide and 560 m long, with the total volume of about $3.8 \times 10^6$ m³. Most materials of the deposit were primarily composed of silty sandstone and limestone that formed from

the Mesozoic era, the Triassic (T). In the area where the collapse occurred, the artificial slope was 7.5 m high with an over 70 degrees angle and covering considerable underlying rocks on the consequent bedding sandstone layer.

### 4.2.3 Toppling collapse

    Toppling failure is one of the most common failure forms of rock deposit slope in the strong

earthquake area. The main failure mode of the toppling failure is bending and overturning, which is caused by bending stress. Toppling generally occurs in steep rocks with vertical joints. Moreover, the soft rock and hard rock interlaced sedimentary rock often occurs toppling failure. When the lower soft interlayer is weathered or eroded by rainfall, the upper loose accumulation body would be suspended, falls, rebounds or rolls down under the action of gravity. Toppling collapse is characterized by

breaking rocks and discontinuous structural cracks, usually triggered by earthquakes or human activities (e.g. hydropower stations building, highways building and other works)(Guo, 2017). Besides, effective inter-granular stress would decrease in deposit material due to the increase in internal


seepage pressure and the decrease in pore water pressure, there by causing a collapse.. This deformation failure model can be defined as toppling collapse.


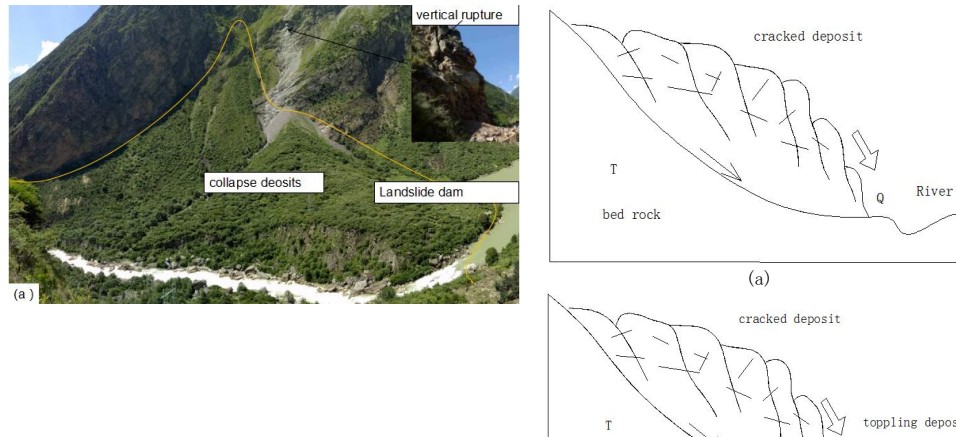

**Fig.10 Aerial photograph and schematic of toppling collapse at Yinping,Mao county, Sichuan, China. (a) aerial photograph at Yinping ;(b)schematic of toppling collapse before failure;(c)schematic of toppling collapse after failure.**

For instance, Yinping toppling collapse was triggered by the 1933 Diexi Ms7.3 earthquake and the 2008 Wenchuan earthquake. This collapse deposits formed from 1993 and blocked the Min river. The geo-structure of this landslide dam is features by the consequent bedding structure and steep cliff.

Because the rock has been falling for 85 years, the collapse deposits are approximately 1000 m wide and 1500 m long, the collapsed rock traveling distance more than 1400 m (Huang, 2009). After the 2008 Wenchuan earthquake, the average thickness of the collapse deposits were over 180 m, and the total volume of were over $2.1 \times 10^8$ m$^3$. Such loose deposits are mostly composed of Quaternary(Q), Triassic metasandstone, crystalline limestone(T) (Fig.10).

### 4.3 Erosion


Erosion often occurred in loose deposit body induced by rainfall or flow in the area with undulating landscape. This mode of motion is usually a spatial continuous motion, and the deposit is carried away by the current from high to low. These processes contributed to the formation of unstable rock and soil masses in the surface of gullies during the different courses of geological erosion (J.Dvorak, 1994),

deformation and destruction, and finally the deposits moved with the grading movement of mud (sand) flow, which depends on the water content, mobility and movement evolution.



### 4.3.1 Scouring and lateral erosion

Scouring and lateral erosion have two main mechanisms: scouring and lateral erosion. River erosion is the direct removal of soil particles by the current. The rate of scouring is determined by the impact of the flow and the erosion resistance of the bank's loose deposit material. When the weight of the upper deposit is greater than the strength of the slip zone, the failure will occur subsequently, resulting in lateral erosion. The process depends on many factors, including the particle composition of the slope material, the water content and the coverage of the vegetation. These two erosion processes are interrelated because the scouring at the bottom of the river bank produces steeper slopes or overhanging clods that are more unstable and may be laterally eroded (Fig.11)

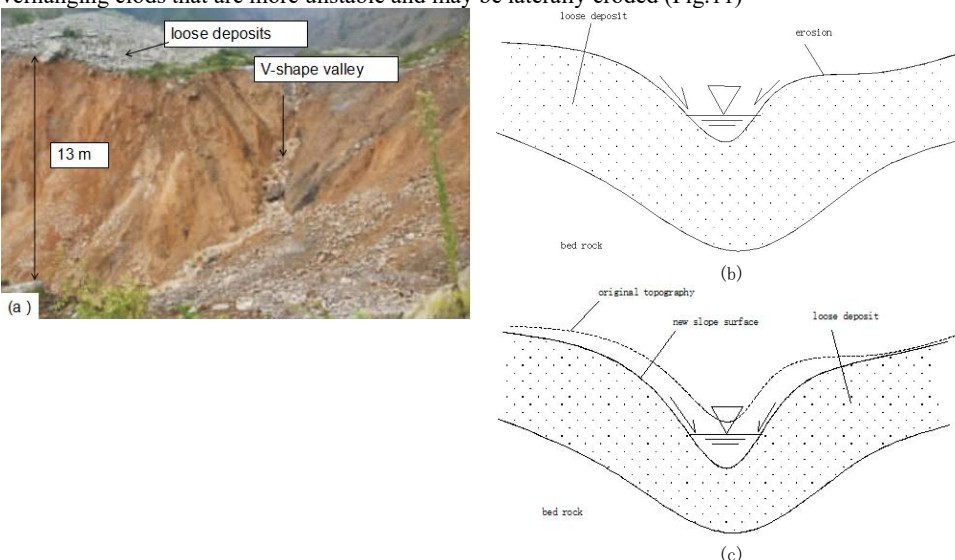

**Fig. 11    Schematic and photograph of Scouring and lateral erosion of loose deposits at Baihe Village, Qingchuan county, Sichuan Province, China,2014. (a)photograph of scouring of Baihe Deposit;(b)schematic of deposits before the failure;(c)schematic of deposits after the failure;**

This type is primarily formed on the surface of loose deposits body, and usually both sides of the slope have U-shaped or V-shaped canyon. They will be strengthened if they occur on a hillside with less vegetation, or both sides of the gullies that have been lost vegetational by earthquake or mining deforestation. Under heavy rain and extreme rainfall conditions, the upstream water continuously washed away the loose deposits, thereby caused the slopes on both sides of the valley to be washed repeatedly, and the valley section gradually expanded and deepened, and finally causes the slope failure (Fig. 11). For instance, the deposits of scouring and later near Baihe Village, Qiangchuan county, Sichuan province, China, 2014, which destroyed 15 houses and caused 3 death, is underlain sericite phyllite of the Silurian system(S). After thousands of years of erosion, the erosion efficiency determines the speed of the material in the collapse process, so the erosion accelerated after the Wenchuan earthquake.





### 4.3.2 Steam Bank erosion

It often occurred at the toe of the loose deposits, and it would be damaged and degraded by the stream, river and lake. Due to the scour, dredging and erosion of curren, the upper part of deposit is not balanced, resulting in local downward cuttiing or collapse of the deformation mode. This study has a typical example for the stream bank erosion of the slope deposit in Soqiao village, Wenchuan County, Sichuan province, China (Fig.12)(Yang,2012).

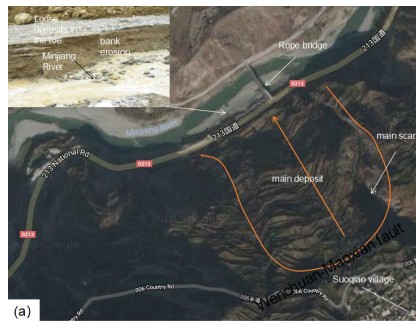

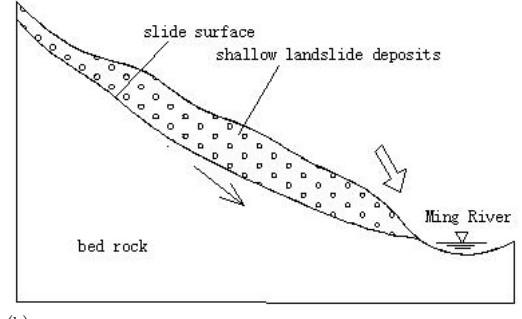

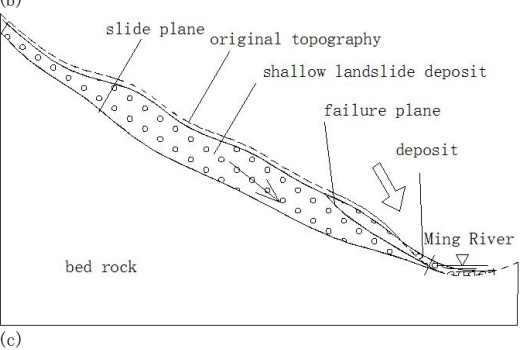

**Fig.12 Schematic and photograph of bank erosion in Rope bridge/Soqiao village, Wenchuan county, Sichuan Province, China,2014. (a)photograph of scouring of bank Deposit;(b) schematic of deposits before the failure;(c)schematic of deposits after the failure.**


Stream bank erosion in Suoqiao village located in the left bank of the Minjiang river, where belong to middle mountain canyon landform. The deposit about 200 m wide and 220-250 m long, while the main body area is approximately $3.88 \times 10^4 \, m^2$ and a total volume of $6.52 \times 10^5 \, m^3$. Most of the material in the toe is gravelly soil, includes $10\% \sim 30\%$ phyllite and limestone debris. The erosion movement

of bank is slow in winter, but the loose deposits travel move faster in rain season. The Suoqiao deposits are unstable because of the bank erosion in the toe and it has a weak sliding surface. Accordingly, it is speculated that landslides will occur in future heavy rain or earthquake conditions.



### 4.3.3 Debris flow cutting

Debris flow cutting typically occurred in a slope of loose deposit body with slope up to 45 degrees,
usually initiated during heavy rainfall, which upstream materials driven by a rainstorm or debris flow.
When the water accumulates rapidly in the upstream, a debris flow will form in the middle and lower
reaches, Subsequently rushed out of the channel, and cut the slope foot result in a steep air surface.
The existence of these loose materials on the slope and the development of heavy rainfall events are
the main reasons for the deformation and failure of these deposits (Xu, 2012).

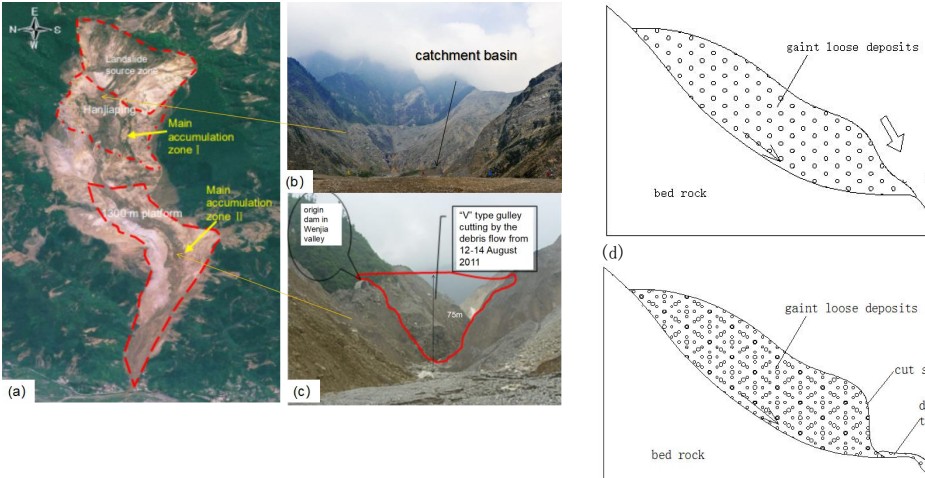

**Fig.13 Schematic and photograph of debris flows cutting in Wenjia gully, Qinping Town, Sichuan Province, China,2010. (a)aerial photograph of Wenjiagou Deposits;(b) photograph in the upstream of debris flow deposits;(c) photograph in the downstream of Wenjiagou debris flow deposits;(d) schematic of the deposits before the failure;(c) schematic of deposits after the failure.**


The famous debris flows cutting type is Wenjia gully which located in the north of Qinping town,
Mianzhu city, Sichuan province, China. The catastrophic deposits were formed by the 2008
Wenchuan earthquake and have been experienced several times of heavy rain and continuous rain.
From September 2008 to September 2011, six large-scale debris flows were formed, which seriously
endangers the safety of life and property of people downstream. The accumulation body has the
relative height difference of 1.49km, the ditch length of 4.9 km, and the overall slope dropped by 306
‰. The accumulation body shows three-level platform accumulation from the profile, with the upper
slope, middle and lower level. The trailing edge and the leading edge of the accumulation body of
Hanjiaping, the first-level platform, are both steep (the gradient is 673.8‰ and 644.4‰ respectively),
which significantly contributes to the formation of the catchment power accelerating the discharge.
The slope falls of the secondary platform (1300 m) and the tertiary platform are relatively small (140.3
‰ and 322.5‰, respectively), whereas the ditch is deep and narrow and the accumulation body




exhibits a large loose thickness, which makes it extremely easy for the erosion and erosion cutting deformation and failure.

**4.4 Flow**

**4.4.1 Debris avalanche**

Debris avalanche was originated from the collapsing material caused by the earthquake. Because of the steep slope, scarce vegetation and extremely loose structure of the deposit, combined with exterior geological force (e.g. aftershocks and human activities), debris flow material in a superficial layer of loose deposit slipped downward with high speed, accompanied by the flow of dust and tumbling sounds of tumbling rocks.

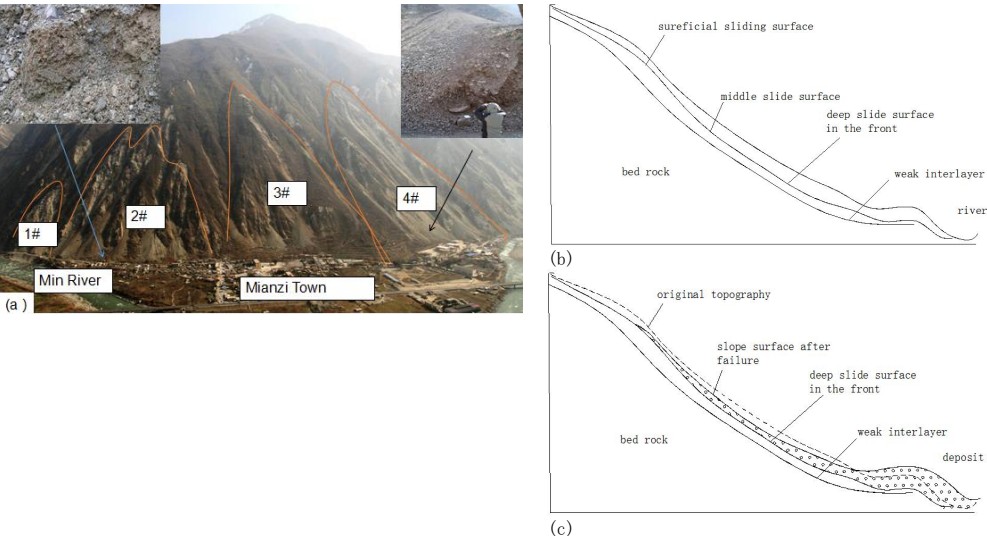

**Fig. 14 Photograph and schematic of debris avalanche at Mengjiacao, Mianzi town, Wenchuan, Sichuan, China. (a)photograph at Mengjiacao; (b)schematic of toppling collapse before failure; (c)schematic of toppling collapse after failure.**

Since 2008, there are hundreds of debris avalanche induced by rainfall or aftershock in Wenchuan earthquake area. The speed of the avalanche chute to the steep channel is usually more than 10 meters per second, whereas some of the landslide flows are much faster. For instance, the Mengjiacao debris avalanche, located in Mianzi town where about 10 km south of Wenchuan county, Sichuan Province, it is a typical avalanche flow in this area. Because of the consequent collapse flow since 2008, the rock or soil has been accumulated in the toe of the slope, and the total volume of this deposits was over $2.5 \times 10^6$ m$^3$. The material of this landslide-debris flows contains characteristic by the loose coarse and fine particles that distributed in different collapse area. The velocity of this landslide-debris in the steep channels usually attain speeds over 12 m/s(Fig.14).


### 4.4.2 Debris flow

Though the number of the debris flow in Wenchuan earthquake area all deposits is a small proportion(1.31%), it has aroused the huge attention from geologists and government because of its

410    fast movement, great harm, difficult prevention and control For instance, the Hongchun gully Debris flow occurred in near the Yinxiu town, Wenchuan county, Sichuan, in 14 August 2010, caused 17 people missing. The debris flow has battered the new 213 National Highway, blocked the Min river, then wiped out Yinxiu town (Fig.15).

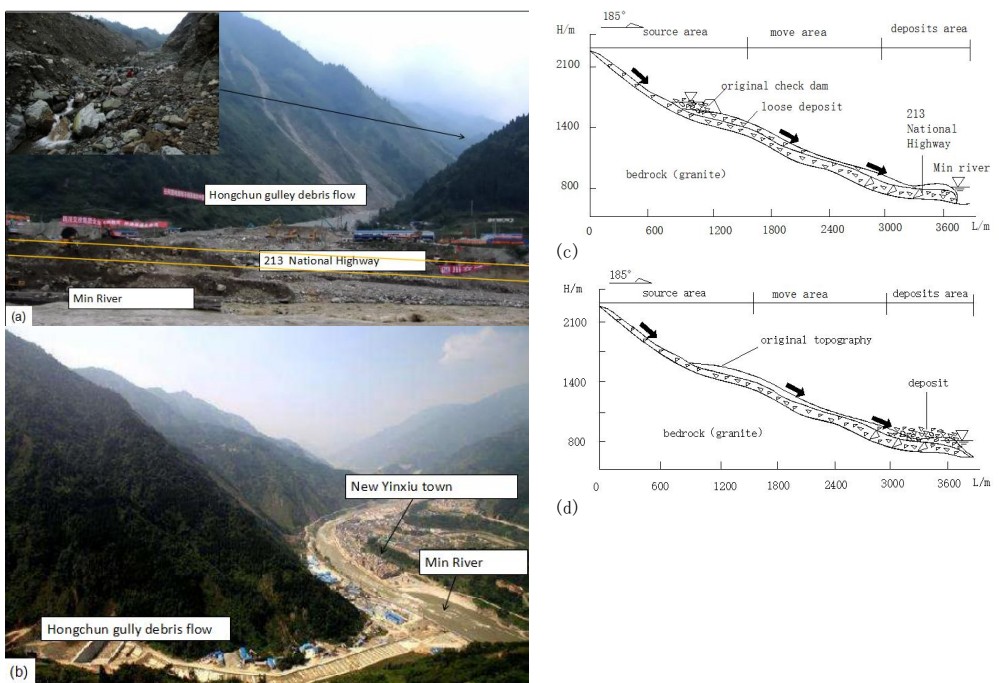

**Fig. 15 Photograph and schematic of debris flow at Hongchun gully, Yinxiu town, Wenchuan, Sichuan, China. (a)photograph at Hongchun gully in 2009; (b) image of Hongchun gully in 2018; (c)schematic of debris flow before failure; (c)schematic of debriws after failure.**

415    Hongchun gully debris flow is one of the 72 debris flows near the Beichuan-Yinxiu Fault in August 2010, which is characteristized by the amount of loose deposits, the steep drop in the shape of gullies and critical rainfall (Tang, 2009). The total volume of this debris flow is nearly $80.5 \times 10^4$ m$^3$, all of these loose materials of the debris flow are composed of granular soil (60%), boulder (25%), rubble (10%) and sand(5%). The channel catchment area covers 3.35 square kilometers, the main channel

420    length is 3.6 kilometers, and the average longitudinal slope of the channel reaches 35. 8%. The top of the slope is 2168.4 m asl, and the gully mouth of debris flow is 700 m asl. The debris flow materials mainly come from three branches in the upper reachsof the Hongchun gully, among which 52    are landslide or collapse deposits, and the total amount of the loose solid material is $3.57 \times 10^6$ m$^3$. Besides,





since the rainfall "8.14" debris flow in Hongcun gully was 16.4 mm per hour and total rainfall reach to
162.1 mm/34 hours before debris flow outbreak, the heavy rainfall is the inducing factor of debris
flow outbreak (Gan, 2012).

**5 Discussion**

Previous studies suggested that different types of accumulation body have significantly different
deformation and destruction mechanism and failure modes (Zhang,2012; Cui, 2014; Huang,2015).
Controlled by various factors (e.g. rock and soil mass structure, geological structure, rainfall and
geographical and geomorphology) of the study area, the accumulation body presents different
deformation and failure modes, and its movement type, speed, scale, geomorphology and landform,
failure modes,etc. are also different(Table 2).

**Table 2 Table of characteristics of deformation and failure of loose deposits in Wenchaun earthquake area**

| Failure type of landslide deposits | | Topography | Material | Travel velocity | Volume | Triggering mechanism |
|---|---|---|---|---|---|---|
| slide | reactivation of old landslide | Mountain, Hill,Talus | Gravel, Sand, Clay, limestone | Various | Small to Huge | Rainfall, Earthquake, Human activities |
| | Slide on weak soil or rocks | Mountain, Hill | Weak rock, Gravel, sand,Silt | Slow | Huge | Rainfall, Earthquake, Human activities |
| | Shallow slide of deep deposits | Mountain, Hill or Valley | Gravelly soils, Weathered rock, | Slow to Ex. Rapid | Small | Earthquake, Weather,Human activities |
| | Integral sliding on bedding rock | Talus, Mountain | Consolidated Soils, Rocks | Slow | Huge | Earthquake, Rainfall, Human activities |
| collapse | Collapse-slide | Mountain | Rock, Soil | Rapid | Small to Huge | Weathering, Rainfall, Earthquake |
| | Cracking sliding of rock collapse | Mountain, Hill | Rock | Slow to Ex. Rapid | Small,Middle | Weathering, Rainfall, Earthquake |
| | Toppling collapse | Steep Cliff | Rock | Rapid | Small to Huge | Weathering, Rainfall, Earthquake |
| erosion | scouring and lateral erosion of deposits | Valley,Gully | Loose Soil or Slay, rock deposits | Slow | Small to Huge | Rainfall, Weather, |
| | Steam bank erosion | Valley, Gully,River | Soils, Sand, Silt | Slow to Ex. Rapid | Small to Huge | Rainfall, Weather |
| | debris flow cutting | Valley,Gully | Rock,Sand | Ex. Rapid | Middle,Huge | Rainfall, Weather |
| flow | Debris avalanche | Mountain | Rock, Clay | Slow to Rapid | Small, Middle | Earthquake, Weather, Rainfall, Rainfall |
| | Debris flow | Mountain, Hill, Valley | Stone, Soil, Sandy gravel | Ex. Rapid | Middle, Huge | |

It is worth noteworthy that topography is a factor significantly affecting the failure of landslide deposit.
It also determines the scale, the shape and the deformation and destruction mode of these
accumulation slopes. Macroscopic topography controls the development and distribution deposit body.
Slopes with different gradients, heights, shapes and vegetation significantly effect the disaster mode of
landslide deposit.
Besides, there was not a clear relationship between the failure mode of the deposits and particle size to
be observed. Deposits are composed of fine particle soil (e.g. sandy soil, gravel soil and clay) that can





be occur sliding, erosion and debris flow. Deposits are composed of the medium and coarse particle

that can also occur such failure as long as there is sufficient rainfall. The precipitation process, rainfall and rainfall intensity significantly effect the formation of debris flow. This study suggests that the continuous rainfall and rainstorm can lead to different failure modes though the same deposits with the same particle size. Vegetation and its root system can weaken and protect the accumulation body erosion from being eroded by rainwater. Investigation statistics reveals that deposit body with

well-developed vegetation primarily formed slip type deformation and destruction, whereas it is unlikely to develop into erosion or collapse. In contrast, collapse or erosion deformation and destruction often occur in places with poor development or underdeveloped vegetation in landslide deposition.

Moreover, the formation of accumulation body was controlled by geological structure. The closer

the distance to Longmen mountain seismic fracture zone, the greater the seismic force and the structure of accumulation body became loose to form debris flow, which may likely be transformed into collapse type and erosion if landslide deposit produced in much closer to fracture zone. Investigation statistics reveals that the failure of landslide deposit in Wenchuan earthquake area was primarily developed in rock and rock-soil (e.g. granite, quartzite, dolomite and limestone). Integral

sliding on bedding rock mostly occurred in rock deposit which is composed of hard rock at the top and weak rock at the bottom. Deposits largely composed of rocks at the top with highly compacted density and weak structural bedding surface, thereby inducing slide on weak soil or rocks easily. Most giant landslide deposits locate in the steep slope near Longmen mountain fault belt, and it was extremely easy to produce catastrophic landslide or debris flow.

**6 Conclusion**

Previous classification studies on loose deposits were based primarily on material, velocity, water content, geotechnical parameters, and other geological hazards, and the effects of topography, landform, volume, and triggering mechanisms are generally not considered. This paper presented a world-recognized classification improvement from the perspectives of topography, velocity, material,

volume and triggering mechanism of loose deposits in strong earthquake area. Thus, the basis of this factors of this classification here are more comprehensive, especially suitable for the actual classification of geological disasters in the meizoseismal, which help to lay a scientific basis for the prevention and control of geological disasters.

According to the results of field investigation and statistical analysis, there were four main types and

12 subcategories of failure modes in loose deposits after 2008 Ms8.0 Wenchuan earthquake area, where are as follows: (1) Slide, covering the reactivation of old landslide, Slide on weak soil or rocks, shallow sliding of deep deposits and integral sliding on bedding rock; (2) Collapse, including



collapse-slide, cracking-sliding rock collapse, topping soil collapse and debris flow cutting; (3) Erosion, e.g. scouring and lateral erosion, steam bank erosion; (4) Flow, e.g. debris avalanche and debris flow.   The investigation statistics on hotspots in Wenchuan earthquake area, Sichuan province, suggests that the failure mode of loosse deposit was mostly of the slide, some of them may occur collapse and erosion, and the fewest of them will occur debris flow.

The category of failure modes in landslide deposits proposed here can serve as preliminary of hazard & risk assessment. More reliable assessment should consider the geotechnical investigation method and means under various conditions, and also rely on accurate geological analysis of landslide deposit. These massive deposits are still highly likely to induce geological disasters under the effect of rainfall, earthquake or human activities. Accordingly, the prediction and stability evaluation of the deformation and damage of loose deposits formed by strong earthquakes remain a matter of great concern.

*Acknowledgements*:This research was partially supported by the First batch of science and technology planning projects of Jiangxi Provincial Department of Education Science and Technology Research Project project (no.GJJ151124) ,Jiangxi province key research and development project (no.20161BBG70051 & no.20171BBG70046), and National natural science foundation of China (no. 41641023 & no.51869012) . We would like to express our gratitude to Prof. Huang Runqiu and Prof. Pei XiangJun in Chengdu University of Technology, whose reviews helped improve this manuscript.

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
