# Peer review of "Failure modes of loose landslide deposits in 2008 Wenchuan earthquake area in China"

_Natural Hazards and Earth System Sciences, 2019_

## Referee Comment (RC1) · Anonymous Referee #1 · 12 Jun 2019

The paper is frequently too generic, descriptive and repetitive. English style is poor, words are often not suitable and several mistakes are present throughout the text. There are many sentences that are not supportd by any kind of proof or data. This cannot be accepted in a scientific paper. The classifications used are not aligned with those widely accepted in the literature, as for example the term "collapse". The discussion is not clear and dows not provide interesting hints.
* * *

---

## Short Comment (SC1) · 16 Jun 2019

The core data (Fig. 1) of slope deposits triggered by the Wenchuan earthquake is totally wrong. There are quite a few papers on landslides (include various types) triggered by the earthquake and the spatial patterns of the landslides is very consistent and recognised. However, the authors neglected most of the important research and represented a self-conceived 2008 Wenchuan earthquake-triggered landslide distribution map but have no correlation with the quake.

---

## Author Comment (AC1) · 17 Jun 2019

Many thanks to Anonymous referee #1! (1) the paper would be modified in detailed and revised according to the opinions of the evaluation experts, focusing on the specific description of cases and avoiding the generalization, description, and repeatability of the paper as much as possible. (2) after polishing by native English experts to provide better English expression, revise case analysis in the article and provide more evidence and data. (3) And refers to the international classification of geological hazards and standardize the classification terms of disasters; (4) modify the discussion to provide some more interesting hint of classifications that are inconsistent with those widely accepted in the literature, such as the word "collapse" change to "rockfall" according to the Varnes classification.

[Figure]

Please also note the supplement to this comment:
https://www.nat-hazards-earth-syst-sci-discuss.net/nhess-2019-25/nhess-2019-25-AC1-supplement.pdf

———————————————————

[Figure]

**Supplement:**

**Failure modes of loose landslide deposits in the 2008 Wenchuan earthquake area in China**

**5 Jianjun Gan1,2, Yi Xia Zhang2**

[revised manuscript text omitted]

I: High mountain plateau region of western Sichuan;11: Shiqu Seda structure denuded hilly plateau are;12: Hongyuan Ruoergai tectonic denuded swampy plains; 13: East bank of Jinsha reive tectonic erosion mountain canyon area; 14: Shapuli mountain erosion or denudation hilly plateau area;15: Yalong river structure erodes the deep valley mountain area;16: Qionlai mountain to Minshan mountain tectonic erosion ridge mountains;17: Gongga mountain structure erodes extremely high mountains;18: Longmen mountain fault erosion slope in the mountain area.

II Mountain area of southwest Sichuan II1: Emei mountain to Wuzhi mount tectonic erosion block mountain area;II2: Xichang Yanyuan Tectonics erodes middle mountainous area of wide valley basin;II3: Liangshan tectonic erosion middle mount area.

III: Mount area in eastern basin in Sichuan; III1: Tectonic erosion low mountain hilly in Sichuan Basin;  $\Pi_1^1$ : Inclined plain sub-region in the front of western fault depression basin;  $\Pi_1^2$ : Mono-clinic low mountain sub-region north of tectonic erosion basin;  $\Pi_1^3$ : Table low hilly sub-region south of erosion tectonic basin;  $\Pi_1^4$ : Parallelism (low mount) valley (hilly) sub-region in eastern of erosion tectonic basin; III2: Michang mount to Dab mount tectonic corrosion bedded middle area; III3: Wu mount to Dalou mount strong karst valley middle mountain area.

**2.2 Seismicity and Rainfall**

105

Several high magnitude earthquake has been recorded in the Longmen Mountain tectonic zone along the eastern margin of Tibetan Plateau (China) in the last few decades. The Ms7.5 Diexi earthquake on August 25, 1933, caused the catastrophic landslide which blocked the Minjiang river and formed three famous "quake lakes". The rock slide depositions had slipped into a channel and formed landslide

dam, then caused deformation and failure, subsequently the water of this lake pour down, and as a result, 2500 people had been killed and more than 6800 houses had been destroyed (Ren, 2017). The

- 110 Wenchuan earthquake on May 12, 2008, and the Lushan earthquake on April 20, 2013, had magnitudes of Ms 8.0 and Ms 7.0, respectively. These epicenters were located Longmen mountain fault, SW-NE of Chengdu City, and the epicenter was located at the depth from 5 to 20 km, within the Eurasian plate of the Yangtze plate.
- The above-mentioned earthquakes occurred in the Longmenshan fault zone, indicating that the strong
  earthquakes in this area are frequent and the geological environment is very fragile, which is the source of power for loose accumulations. These recurring earthquakes are the result of the relative uplift of the Tibetan Plateau and the relative decline of the Sichuan Basin. The relative movement of the Qinghai-Tibet Plateau and the Sichuan Basin resulted in the uplift of the Longmen Mountains and formed a large seismic zone parallel to the eastern margin of the Qinghai-Tibet Plateau. The
  Longmenshan fault zone includes three major fractures, namely Maoxian-Wenchuan fault and
- Yingxiu-Beichuan fault and Pengxian-Guanxian fault, which are widely distributed on the two largest anticlinorium, i.e. the Pengguan anticlinorium and Baoxing anticlinorium. Due to the violent new tectonic movement in the area, the rock mass is broken and the earthquake is frequent, causing a large number of loose deposits (Fig. 2). As shown in Fig. 2, three major faults were formed, i.e, the fault
- 125 located at the junction of Longmen Shan and Sichuan basin, the piedmont fault, also known as Pengxian-Guanxian Fault, which is approximately parallel to the Longmen mountain, and the 240 km main central fault, which is also known as Yingxiu-Beichuan Fault; and the Maoxian-Wenchuan Fault, also known as the Maoxian-Wenchuan Fault.
- Most typical loose deposits triggered by the earthquake occurred in Longmen Mountain of Wenchuan, which is around 60 km of Chendu city, Sichuan Province, near the east fringe of Tibetan plateau, China (Fig.1). Based on the multi-source remote sensing data and field survey data from 2009 to 2018 provided by the China Geological Survey (CGS), rainfall is the main cause of landslides, rock-avalanches, and mudslides caused by loose debris deformation. Among them, the period of 2010-2014 is the peak of the development of rainfall and geo-hazards, and hundreds of geological
- disasters were caused by the failure of loose deposits after the 2008 Wenchuan earthquake.

---

## Author Comment (AC2) · 17 Jun 2019

Many thanks to Dr. Xu for his valuable comments. The data in this paper is based on the statistical analysis of geological hazard data in Sichuan after the 208 Wen earthquake by China Geological Suvey. As Dr. Xu said, the data of these geological hazard accumulations are not entirely caused by the earthquake but there in the earthquake area. Some geological disasters are indeed unrelated to earthquakes.

---

## Referee Comment (RC2) · Anonymous Referee #2 · 31 Jul 2019

The manuscript deals with a potentially very interesting topic regarding mass wasting processes involving loose deposits of the earthquake-prone Wenchuan area (China) which, also due to geological features, mountainous morphology and intense rainfall regime, is a geomorphologically active area, as it is clearly testified by various landslide and erosion phenomena occurring in it. Notwithstanding the general interesting topic, the scientific quality of the manuscript is poor due to the complete lack of a clear scientific focus, if not novel. In fact, seeming the main focus being addressed to the creation of a new classification for failures modes of loose landslide deposits, the following weakness points are critical: 1) Were all deposits formed by pre-existing landslide phenomena or by other erosional processes. In the first case, landslide processes will be of reactivation type only. 2) Landslide and erosional processes are wrongly mixed

and, in some cases, linked to the same classification (e.g. Cruden & Varnes, 1996). 3) Being the Wenchuan area no figure regarding isoseismal map or historical distribution of earthquakes is shown. 4) Geotechnical data is declared to have been used but any elaboration of it, even simple, was not shown. Moreover, the most important literature concerning landslide classification has not been clearly applied to analyze and interpret mechanisms of phenomena studied or not well considered (e.g. Hungr et al., 2001 regards the flow-like landslide only). According to these major observations, and not considering specific comments concerning several inconsistencies throughout the text, this Reviewer considers the manuscript far beyond the acceptance limit of NHESS journal.

---

## Short Comment (SC2) · 2 Aug 2019

Explanation of the Revision

Dear Editors and Reviewers: Thank you to tell me Anonymous Referee#2's valuable advice from public review and discussion. According to the opinions and suggestions of the reviewer, the main proceedings as flowing:

Response to comment: 1) Were all deposits formed by pre-existing landslide phenomena or by other erosional processes. In the first case, landslide processes will be of reactivation type only. Response: Based on the all deposits formed by pre-existing landslide phenomena or other erosional processes, in the first case , the "4.1.1 Reactivation of old landslide" has been changed to"4.1.1 Rotational of the loose deposit".

Response to comment: 2) Landslide and erosional processes are wrongly mixed and, in some cases, linked to the same classification (e.g. Cruden & Varnes, 1996). Response: Sliding and erosion are classified according to Cruden & Varnes' classification in 1996. E.g. "4.1.4 Integral sliding" in the original text is changed to "4.1.4 Translational slide"; and "4.3.1 scouring and lateral erosion" has been changed to "4.3.1 Sheet erosion" and the "4.3.2 Steam bank erosion" changed to "Gully erosion"; and " 4.4.1 debris avalanche" has been changed to "4.4.1 rock avalanche," all the modification have already amended the corresponding contents of the article.

Response to comment: 3) Being the Wenchuan area no figure regarding isoseismal map or historical distribution of earthquakes is shown. Response: The isoseismal map has been added in Fig.2.

Response to comment: 4) Geotechnical data is declared to have been used but any elaboration of it, even simple, was not shown. Moreover, the most important literature concerning landslide classification has not been clearly applied to analyze and interpret mechanisms of phenomena studied or not well considered (e.g. Hungr et al., 2001 regards the flow-like landslide only). Response: 4) limited to the number of words in the article, the geotechnical data has not been included, and the relevant content has been deleted. The other types of landslides provided by Hungr et al. 2001 were not included due to the purpose of this study is to develop a classification method of post-earthquake loose deposit, We will conduct further research in the future.;

Kind regards.

Jianjun GAN

Email: ganjianjun@nit.edu.cn

Please also note the supplement to this comment:
https://www.nat-hazards-earth-syst-sci-discuss.net/nhess-2019-25/nhess-2019-25-SC2-supplement.zip

---

## Author Comment (AC3) · 14 Sep 2019

Reviewer C (Anonymous Referee #2 ) Comments from Referees: The manuscript deals with a potentially very interesting topic regarding mass wasting processes involving loose deposits of the earthquake-prone Wenchuan area (China) which, also due to geological features, mountainous morphology and intense rainfall regime, is a geomorphologically active area, as it is clearly testified by various landslide and erosion phenomena occurring in it. Notwithstanding the general interesting topic, the scientific quality of the manuscript is poor due to the complete lack of a clear scientific focus, if not novel. Response: The loose accumulation in Wenchuan earthquake areas often leads to catastrophic events, such as on August 20, 2019, when heavy rains left another 12 dead and 26 missings. Therefore, it is of great significance to study the de-

formation and failure modes of post-earthquake loose accumulation. Through the case study of typical post-earthquake loose accumulation bodies, this paper summarizes 12 kinds of common failure types. We focus on the classification of the deformation and failure of the loose deposits in the post-earthquake area. Comments from Referees: In fact, seeming the main focus being addressed to the creation of a new classification for failures modes of loose landslide deposits, the following weakness points are critical: 1) Were all deposits formed by pre-existing landslide phenomena or by other erosional processes. In the first case, landslide processes will be of reactivation type only. Response: Based on the all deposits formed by pre-existing landslide phenomena or other erosional processes, in the first case, the "4.1.1 Reactivation of old landslide" has been changed to"4.1.1 Rotational of the loose deposit" at Page 9 Line 199 and Table 2. 2)Landslide and erosional processes are wrongly mixed Interactive comment Printer-friendly version Discussion paper and, in some cases, linked to the same classification (e.g. Cruden & Varnes, 1996). Response: Sliding and erosion are classified according to Cruden & Varnes' classification in 1996. E.g. "4.1.4 Integral sliding" in the original text is changed to "4.1.4 Translation on bedrock" at Page 12 Line 261; and "4.3.1 scouring and lateral erosion" has been changed to "4.3.1 Sheet erosion" in Page 17 Line 353; and the "4.3.2 Steam bank erosion" changed to "4.3.2 Gully erosion" in Page 18 Line 375; and "4.4.1 debris avalanche" has been changed to "4.4.1 rock avalanche" on Page 21 Line 414; all the modification have already amended the corresponding contents of the article.

3)Being the Wenchuan area no figure regarding isoseismal map or historical distribution of earthquakes is shown. Response: 3) Being the Wenchuan area no figure regarding isoseismal map or historical distribution of earthquakes is shown. Response: The isoseismal map has been added in Fig.2 at Page 5 Line 122.

4)Geotechnical data is declared to have been used but any elaboration of it, even simple, was not shown. Moreover, the most important literature concerning landslide classification has not been clearly applied to analyze and interpret mechanisms of

phenomena studied or not well considered (e.g. Hungr et al., 2001 regards the flow-like landslide only). Response: 4) Because the research focuses on deformation and failure mode, this paper mainly expounds the influence of formation lithology on deformation and failure of loose accumulation body in the case study. Other geotechnical data such as formation thickness, particle size distribution, and mechanical parameters are only for reference. The other types of landslides provided by Hungr et al. 2001 were not included due to the purpose of this study is to develop a classification method of post-earthquake loose deposit, 5 subclassification has been modified in Table 2 based on the Topography, Material, Travel velocity, Volume, and Triggering mechanism at Page 23 Line 460. All changes are shown in blue or red font.

Please also note the supplement to this comment:
https://www.nat-hazards-earth-syst-sci-discuss.net/nhess-2019-25/nhess-2019-25-AC3-supplement.pdf

---

## Author Comment (AC4) · 14 Sep 2019

**Reviewer A (Anonymous Referee #1)**

**Comments from Referees:** The paper is frequently too generic, descriptive and repetitive. English style is poor, words are often not suitable and several mistakes are present throughout the text.

**Response:** This article focuses on the general rules of deformation and failure of loose accumulation slope in Wenchuan postearthquake area, a large number of cases were discusses and analyzes base on the field investigation to provide a reference for engineers and peers. Some cases have been modified in detailed and revised according to the opinions of the reviewers to avoiding the generalization, description, and repeatability of the paper. All the paper has been edited by the local English speaker.

**Comments from Referees:** There are many sentences that are not supportd by any kind of proof or data. This cannot be accepted in a scientific paper.

**Response:** after polishing by native English experts to provide better English expression, revise case analysis in the article and provide more evidence and data to suit for a scientific paper in Page 3 Line 70, Page 8 Line 172-174, Page 12 Line 253-254, etc.

**Comments from Referees:** The classifications used are not aligned with those widely accepted in the literature, as for example the term "collapse".

**Response:** And refers to the international classification of geological hazards and standardize the classification terms of disasters; Such as the term "Collapse" has been modified to "Rockfall", and "Collapse-slide" changed to "rockfall-slide", and "scouring and lateral erosion of deposits" has been changed to "Sheet erosion", and "Steam bank erosion" has been changed to "Gully erosion" etc. according to the Varnes classification in Table 2, at Page 23 Line 460.

**Comments from Referees:** The discussion is not clear and does not provide interesting hints.

Response: (4) modify the discussion to provide some more interesting hint of classifications that are inconsistent with famous catastrophic geohazards like the Wenjian gully debris flow in Page 20 Line 399-412, Hongchun gully debris flow in Page 22 Line 439-450.

---

## Author Comment (AC5) · 14 Sep 2019

**Reviewer B (Chong Xu, 16 Jun 2019 )**

**Comments from Referees:** The core data (Fig. 1) of slope deposits triggered by the Wenchuan earthquake is totally wrong. There are quite a few papers on landslides (include various types) triggered by the earthquake and the spatial patterns of the landslides is very consistent and recognised. However, the authors neglected most of the important research and represented a self-conceived 2008 Wenchuan earthquake-triggered landslide distribution map but have no correlation with the quake.

**Response:** The core data (Fig.1 ) of slope deposits triggered by the Wenchuan earthquake has been changed to "Statistical distribution of Loose deposits postearthquake in Sichuan Province, China" in Page 4 Fig.1.

In this paper is based on the statistical analysis of geological hazard data in Sichuan after the 2008 Wenchuan earthquake by China Geological Survey. As Dr. Xu said, the data of these geological hazard accumulations are not entirely caused by the earthquake but there in the earthquake area. This manuscript mainly studies the deformation and failure modes of loose accumulation bodies in Sichuan province after the earthquake. Therefore, all the loose accumulation bodies in this region are taken as research objects.